# Atrial Natriuretic Peptide Affects Skin Commensal *Staphylococcus epidermidis* and *Cutibacterium acnes* Dual-Species Biofilms

**DOI:** 10.3390/microorganisms9030552

**Published:** 2021-03-08

**Authors:** Maria Alekseevna Ovcharova, Olga Vyacheslavovna Geraskina, Natalya Dmitrievna Danilova, Ekaterina Alexandrovna Botchkova, Sergey Vladislavovich Martyanov, Alexey Valeryevich Feofanov, Vladimir Konstantinovich Plakunov, Andrei Vladislavovich Gannesen

**Affiliations:** 1Laboratory of Viability of Microorganisms, Federal Research Center “Fundamentals of Biotechnology” of Russian Academy of Sciences, 117312 Moscow, Russia; masha_ovcharova_97@mail.ru (M.A.O.); leo_523@mail.ru (N.D.D.); semartyan@inbox.ru (S.V.M.); plakunov@inmi.ru (V.K.P.); 2Department of Bioengineering, Faculty of Biology, Lomonosov Moscow State University, 119234 Moscow, Russia; olgsamsonova@yandex.ru (O.V.G.); avfeofanov@yandex.ru (A.V.F.); 3Laboratory of Microbiology of Anthropogenic Habitats, Federal Research Center “Fundamentals of Biotechnology” of Russian Academy of Sciences, 117312 Moscow, Russia; botchkovaekat@gmail.com; 4Shemyakin-Ovchinnikov Institute of Bioorganic Chemistry, Russian Academy of Sciences, 117997 Moscow, Russia

**Keywords:** biofilms, dual-species biofilms, atrial natriuretic peptide, hormones, skin microbiota, *Staphylococcus epidermidis*, *Cutibacterium acnes*

## Abstract

The first evidence of the atrial natriuretic peptide (ANP) effect on mono-species and dual-species biofilms of skin commensals *Cutibacterium acnes* and *Staphylococcus epidermidis* was obtained in different model systems. Elucidation of the mechanism of action of hormones on the microbial communities of human skin is an important physiological and medical aspect. Under anaerobic conditions, ANP at a concentration of 6.5 × 10^−10^ M inhibits the growth of *S. epidermidis* biofilms and stimulates the growth of *C. acnes* biofilms, and a lesser effect has been demonstrated on planktonic cultures. In biofilms, ANP stimulates aggregation in *C. acnes* and aggregate dispersion of *S. epidermidis*, while in *S. epidermidis*, ANP also stimulates the metabolic activity of cells. Analysis of dual-species biofilms has shown the dominance of *S. epidermidis*, while ANP increases the ratio of *C. acnes* biomass in the community. ANP decreases the growth rate of *S. epidermidis* biofilms and increases that of *C. acnes*. The effect of ANP is not dependent on the surface type and probably affects other targets in microbial cells. Thus, the potential regulatory effect of human ANP on skin microbe dual-species communities has been shown, and its potential has been demonstrated to change microbiota homeostasis on the skin.

## 1. Introduction

Interactions between the human humoral systems and the human microbiota are of increasing interest to microbiologists, cosmetologists and dermatologists all around the world. New methods of skin disorders treatment and novel cosmetics development has become more focused on skin microbiota homeostasis and its interactions with the human organism. In cosmetology and dermatology, novel approaches can be developed on the basis of the host humoral system—its modulation and homeostasis shifts. Recently, regulatory molecules such as hormones were shown to have an impact on bacteria in comparison with other “classic” antimicrobial molecules such as defensins [1]. The systematic study of hormones began in the early 2000s with the works of Lyte and coauthors [2]. Regarding hormones, catecholamines are the most studied hormones as bacterial affectors in comparison with other hormones, and natriuretic peptides are poorly investigated. Natriuretic peptides (NUPs) are a family of compounds synthesized in the human body that are responsible for the regulation of cardiovascular homeostasis and osmoregulation [3]. In addition, they perform numerous other functions in humans [4]. The three main types of NUPs are atrial or A-type natriuretic peptide (ANP), B-type natriuretic peptide (BNP) and C-type natriuretic peptide (CNP). ANP and BNP are produced by cardiomyocytes, and endothelial cells are the source of CNP [4]. Despite the importance of NUPs for eukaryotic human cells and their apparent specificity for human cell targets, NUPs were recently shown to have the ability to affect prokaryotic cells, at least in the bacterial domain. The effect of NUPs on microbial growth, virulence and biofilm formation was shown and confirmed but has not yet been intensely studied. There is some information on the relationship between NUPs and the gram-negative skin opportunistic pathogen *Pseudomonas aeruginosa*, which is the most studied compared to other microorganisms. CNP increased the toxicity and virulence of *P. aeruginosa* but had no impact on its growth [5,6]. CNP inhibited biofilm formation of *P. aeruginosa* in a dose-dependent manner. A bacterial sensor for CNP, AmiC, was described in *P. aeruginosa*. Moreover, it was shown in silico that AmiC and NPR-C, preferential CNP receptors in humans, have a similar 3D structure. AmiC can bind CNP and isatin, an antagonist of NPR [7]. However, it was recently shown that in addition to AmiC, CNP might have multiple modes of action on *P. aeruginosa* depending on the concentration [8].

The effects of ANP and CNP were studied on two strains of human skin commensal microorganisms, *Staphylococcus epidermidis* MFP04 and *Staphyloccus aureus* MFP03. Both NUPs affected biofilms and, to a lesser degree, planktonic cultures. The development of *S. epidermidis* biofilms was stimulated at 37 °C and suppressed at 33 °C, while in the case of *S. aureus*, the effect was reversed: stimulation at 33 °C and inhibition at 37 °C [9]. The effect of NUPs was also dose dependent.

NUPs were tested on binary biofilms of two staphylococci, *S. epidermidis* MFP04 and *S. aureus* MFP03 [9]. In dual-species biofilms at 37 °C, ANP and CNP increased the competitiveness of *S. epidermidis* against *S. aureus*. In another study, binary biofilms of *S. aureus* MFP03 and *Cutibacterium acnes* RT5 were strongly affected by ANP and CNP depending on the aero-anaerobic conditions and temperature. *C. acnes* became more competitive against *S. aureus* in the presence of both studied NUPs [10]. These results are of special interest since *S. aureus* is considered to be more pathogenic and potentially troublesome than *C. acnes* [11]. Thus, NUPs can potentially be an additional factor in maintaining human microbiota homeostasis.

*S. epidermidis* and *C. acnes* are gram-positive nonmotile bacteria of the Firmicutes and Actinobacteria phyla, respectively, that play a major role in skin homeostasis [12]. The importance of *S. epidermidis* and *C. acnes* as parts of the human cutaneous microbiota is undoubtable because these bacteria are some of the most widespread and abundant skin commensals [13]. The positive role for these species is known, as they are the major agents for maintaining skin homeostasis and fighting against pathogens [12]. However, they can also switch into opportunistic pathogen lifestyles and be causative agents of skin diseases. Their simultaneous presence inside acne lesions is well known [14,15,16], and these bacteria can also be isolated together from other inflammation sites, such as endodontic lesions [17], where these bacteria can be the most abundant species. To date, interactions of these bacteria inside the dual-species community have been shown to be antagonistic in some cases. For instance, *C. acnes* is able to produce short fatty acids that are able to inhibit *S. epidermidis* growth [18]. On the other hand, *S. epidermidis* is able to inhibit *C. acnes* via the production of potential bacteriocin-like compounds [19] or to facilitate inflammation quenching via interaction with keratinocytes and modulation of toll-like receptors and interleukins by synthesizing succinic acid and other compounds [20]; hence, *S. epidermidis* potentially has the ability to quench acne vulgaris. There is some evidence of the possibility of using *S. epidermidis* as a probiotic against *C. acnes* [21]. In this way, it is interesting to know how human regulatory molecules can potentially affect the community of *S. epidermidis* and *C. acnes* on the skin and how the community can be affected by human hormones in different physiological states. In this work, we tested ANP as a potentially active compound on mono-species and dual-species biofilms of *C. acnes* and *S. epidermidis* using techniques described in our previous studies [9,10].

## 2. Materials and Methods

### 2.1. Bacterial Strains and Cultivation

The *Staphylococcus epidermidis* ATCC14990 strain was stored at room temperature (RT) in tubes with semiliquid lysogeny broth (LB, Dia-M, Moscow, Russia) supplemented with 0.4% agar covered with sterile mineral oil. For experiments, a portion of biomass was plated on reinforced clostridial medium (RCM) with 1.5% agar and was cultivated at 33.5 °C for 24 h. The RCM contained the following (g/L): peptone (Dia-M)–10, yeast extract (Dia-M)–13, glucose (Dia-M)–5, NaCl (Dia-M)–5, sodium acetate (Dia-M)–3, starch (Dia-M)–1, cysteine-HCl (BD)–0.5, distilled water, pH 6.8–7.0. To prepare overnight cultures, a single colony from the Petri dish was inoculated into a 50 mL Erlenmeyer flask with 15 mL of sterile LB and incubated at 150 rpm and 33.5 °C to obtain an optical density (OD) at 540 nm of 1.7–2.0. *Cutibacterium acnes* HL043PA2 with the acneic RT5 ribotype was stored under lyophilized conditions in Nalgene^®^ cryotubes (Merck, Darmstadt, Germany) filled with CO_2_ and sealed with parafilm at RT. A portion of lyophilized culture powder was inoculated into sterile tubes with plastic screw plugs filled fully with liquid RCM, and the culture suspension was incubated at 33.5 °C under static conditions for 3–5 days. Then, the suspension was plated onto RCM agar to obtain single colonies. For experiments, a single colony was inoculated into tubes with RCM prepared as described above and incubated for 72 h at 33.5 °C to a final OD_540_ of 1.8–2.0. In experiments, cultures were adjusted to a standard OD_540_ appropriate for each experiment (0.5, 1, 2 or 4) using sterile physiological saline (0.9% NaCl in distilled water) as described below. No pre-treatment of cultures with ANP was performed. No ANP was added to initial cultures.

### 2.2. Natriuretic Peptides

Human ANP (Alfa Aesar, Ward Hill, MA, USA) has a molecular weight of 3080.47 g/mol. The peptide was reconstituted in Milli-Q water and stored dissolved at −20 °C with a molar concentration of 1.623 × 10^−4^. According to data in the literature on the physiological NUP concentrations in human blood plasma [22,23], we studied the effect of the peptide at concentrations ranging from 6.5 × 10^−12^ to 6.5 × 10^−9^ M. Because there is no information about how much ANP is actually present in skin glands, we used its concentration in the blood (6.5 × 10^−12^ M), and we also tested a range of increased concentrations, as occurs during some pathologies. ANP was added to the experimental medium immediately before the start of the experiment. 

### 2.3. Mono-Species Biofilm Growth on PTFE Cubes

In experiments, both microbes were cultivated in anaerobic conditions to simulate the microniche on skin appropriate for *C. acnes*. The model with polytetrafluoroethylene (PTFE) cubes is a robust system that allows easy analysis of biofilm and planktonic culture growth simultaneously. Additionally, the surface area of the cubes (~20 cm^2^) in a test tube corresponded approximately to the area of eight wells of a standard 96-well plate, providing stability and reliability of the experimental results; thus, there are fewer statistical fluctuations in comparison with the traditional microtiter plate method [24]. Biofilms on PTFE cubes were grown as described previously [24] with modifications. Briefly, 21 chemically pure PTFE cubes of size 4 × 4 × 4 mm were placed in glass tubes of the standard volume of 22 mL with screw plugs. After 10 mL of RCM was added to each tube, tubes were shut loosely with screw plugs and were autoclaved at 112 °C. After sterilization and cooling, 11 mL of RCM was added to the tubes for anaerobic cultivation to fill them fully. Tubes without ANP addition were used as a positive control. Then, 350 μL of prepared cell suspension with OD_540_ = 0.5 was added to each tube, and at least two tubes were used as negative controls without bacterial inoculation. Tubes were incubated at 33.5 °C with shaking at 180 rpm for 72 h to obtain mature biofilms. *S. epidermidis* biofilms were also analyzed after 24 h. After incubation, the OD_540_ was measured with the use of empty controls without bacterial inoculation, and biofilm crystal violet (CV) staining was subsequently performed to analyze the total amount of biofilms on the PFTE surface. Biofilms were stained as described previously [24]. Briefly, cubes were washed twice gently with RT tap water to remove planktonic culture and fixed with 3 mL of 96% ethanol for 20 min. After fixation, ethanol was removed, cubes were dried and 2 mL of the 0.5% CV solution in distilled water was added to each tube and incubated for 20 min at RT. Next, the CV was removed, and cubes were washed 6 times gently with RT tap water and placed into new clear glass tubes to be covered with 3 mL of 96% ethanol for dye extraction. The OD_590_ was measured after 30 min of extraction with the use of empty controls.

### 2.4. Biofilm Growth on Glass Microfiber Filters

Both microbes were cultivated in anaerobic conditions as in previous experiments. Mono-species and dual-species biofilms of *S. epidermidis* and *C. acnes* were grown on glass microfiber filters of grade GF/F that were 21 mm in diameter (GMFF, Whatman, Little Chalfont, UK). Bacterial suspensions were adjusted to OD_540_ = 0.5 and 1. The sterile GMFFs were placed onto RCM agar in Petri dishes with the addition of ANP, and the controls did not include the addition of the hormone. To obtain mono-species biofilms, an OD_540_ = 0.5 suspension was used. For dual-species biofilms, OD_540_ = 1 suspensions of each species were mixed in equal proportions. After 15 μL of the appropriate culture was inoculated onto the center of the GMFF, Petri dishes with GMFFs were incubated in anaerobic conditions in GasPak containers (BD, Franklin Lakes, NJ, USA) equipped with Anaerogaz gas-generating cartridges (NIKI-MLT, Saint-Petersburg, Russia) at 33.5 °C for 24 h or 72 h to analyze the dynamics of dual-species biofilm growth in comparison with mono-species biofilms. Hence, all the incubation time bacteria were exposed to ANP. After incubation, some filters with biomass were disrupted to count colony forming units (CFUs) and evaluate aggregation. A filter was placed into a tube with 10 mL of sterile PS, disrupted with a sterile glass stick and vortexed for 1 min to make a homogenous suspension. Aggregation analysis was performed using light microscopy (Carl Zeiss Jena, Oberkochen, Germany) of CV-stained samples after biofilm dispersion as described above. Ten microliters of the suspension was dropped onto a glass slide, fixed and stained with CV to analyze cell aggregation after biofilm dispersion. Samples were analyzed first by visual evaluation. Then, at least 5 photos of the most representative viewing fields were taken using the 90× oil immersion objective and the microscope-associated camera (ToupView Photonics, Hangzhou, China). The number of single cells, aggregates and aggregate sizes (cells) were analyzed. For CFU count, 15 µL of prepared and diluted suspension was dropped onto Petri dishes with RCM agar and spread with a spatula. Colonies were obtained after incubation in an anaerobic atmosphere at 33 °C for 7 days. Another part of the filters was stained with 3-(4,5-dimethyl-2-thiazolyl)-2,5-diphenyl-2-tetrazolium bromide (MTT) as described previously [9]. Briefly, MTT is an acceptor of electrons from electron transport pathways; when reduced, it transforms to formazan, which is not soluble in water. The amount of formazan generated is proportional to the metabolic activity of the bacteria.

### 2.5. Biofilm Growth for Confocal Laser Scanning Microscopy (CLSM)

Both microbes were cultivated in anaerobic conditions as in previous experiments. To analyze the biofilm architecture more deeply and provide a more detailed study of ANP effects on species interactions in intact dual-species biofilms, Confocal Laser Scanning Microscopy (CLSM) of mono-species and dual-species biofilms of *C. acnes* and *S. epidermidis* was performed. The goal of the experiments was to analyze parallel samples of mono-species biofilms hybridized with an appropriate fluorescent probe and samples of dual-species biofilms. For the dual-species samples, *S. epidermidis* was stained in the first half of the samples, and in the other half of the samples, *C. acnes* was stained. Biofilms were obtained in 24-well black plates with flat glass bottoms (Eppendorf, Hamburg, Germany). Cultures of *S. epidermidis* and *C. acnes* of OD_540_ = 1 and 2 and their dual-species mixture were prepared as described above. One milliliter of RCM with or without ANP was inoculated in each well, and 17 µL of the appropriate culture was inoculated into the well. Biofilms were grown in anaerobic conditions using the GasPack-Anaerogaz system for 24 and 72 h. After incubation, the cell suspension was gently removed, and biofilms on the bottom were washed twice with sterile PS. Then, the samples were stained using a fluorescent in situ hybridization technique.

### 2.6. Fluorescent In Situ Hybridization (FISH) of Dual-Species Biofilms

Specific Fluorescent In Situ Hybridization (FISH) probes were chosen for *S. epidermidis* and *C. acnes* to perform hybridization. Before the experiments, false hybridization tests were performed to assure the absence of false-positive hybridization. For *S. epidermidis*, the probe 5′-TCCTCCATATCTCTGCGC-3′ [25] was labeled with fluorescein amidite (FAM) on the 5′ end. The hybridization procedure was performed as described previously [25]. Briefly, *S. epidermidis* mono-species and dual-species biofilms prepared in a 24-well microplate as described above were fixed with 96% ethanol for 20 min at RT. Then, biofilms were pretreated for 15 min with lysozyme (1 mg/mL in 10 mM Tris-HCl pH 8.0) and for 5 min with lysostaphin (10 μg/mL in 10 mM Tris-HCl pH 8.0). Then, the cell membranes were permeabilized by exposing them to increasing concentrations of ethanol (50, 80, and 100%) in pH 7.4 phosphate buffer at room temperature for 3 min. Probes for the detection of air-dried samples were covered with 200 μL of a hybridization buffer (0.9 M NaCl; 0.02 M Tris-HCl pH 8.0; 0.01% SDS, 20% formamide) containing 5 μg/mL of the probe and incubated at 46 °C in a humid chamber. After 90 min, slides were transferred to washing buffer (0.225 M NaCl, 0.02 M Tris-HCl, 0.01% SDS) and incubated for 15 min at 48 °C. Then, the samples were dried at RT and covered with Prolong Gold antifade mountant (Thermo Fisher, Massachusetts, U.S.) to prevent fast signal attenuation. Then, the plates were covered with aluminum foil and kept in the dark at 4 °C for at least 24 h for mountant fixation.

For *C. acnes*, the probe 5′-GCCCCAAGATTACACTTCCG-3′ [26] was labeled on the 5′ end with rhodamine R6G. FISH was performed as described previously [10,26]. Briefly, biofilms in wells were fixed with 96% ethanol for 30 min at room temperature. Subsequently, ethanol was removed, and biofilms were dried at RT. Fixed biofilms were treated at 200 mL per well with a lysozyme solution (0.1 mg/mL in buffer 0.1 M Tris–HCl pH 8.0 C 0.05 M EDTA). Lysozyme treatment was conducted at 37 °C for 3 h in a humid chamber. Subsequently, the lysozyme solution was removed, and 200 mL of the lysostaphin solution (10 μg/mL in 10 mM Tris-HCl pH 8.0) was added to each well. Plates were incubated in the same system for 10 min at 37 °C. Then, the lysostaphin solution was removed and wells were washed twice with phosphate buffer (pH 7.4). Cell membranes were permeabilized by exposing them to increasing concentrations of ethanol (50, 80, and 100%) in pH 7.4 phosphate buffer at RT for 3 min. Then, the samples were dried at RT and covered with Prolong Gold antifade mountant as described above.

CLSM was performed on an LSM510 confocal microscope (Carl Zeiss, Germany) equipped with a C-Apochromat 63×/1.2 W corr objective. FAM-labeled samples were excited with a 488 nm wavelength Ar^+^ laser. Fluorescence was detected at wavelengths longer than 505 nm using the LP505 longpass filter. R6G-labeled probes were excited with a 543 nm wavelength He,Ne laser, and fluorescence was detected at wavelengths longer than 560 nm using an LP560 longpass filter. In both cases, the diameter of the confocal diaphragm corresponded to one Airy disk. The lateral resolution of the images was approximately 250 nm, and the optical section thickness was <0.9 μm. The scanning step in the Z-direction was 1 µm for routine scanning and 0.5 µm for more detailed 3D images. To increase the reliability of the results, microscopy was conducted in a blinded manner.

The visualization of images was conducted in Zen Blue Edition (Carl Zeiss, Germany), and the numeric processing of files was performed in ImageJ software (NIH, Bethesda, MD, USA) using the Comstat2 plugin. The average biomass density (µm^3^/µm^2^) was calculated for each image using the threshold 15, the ratio of biomass in dual-species biofilms was evaluated and the effect of ANP in immature and mature biofilms was measured.

### 2.7. Kinetics Measurements of Bacterial Growth

Sterile 96-well flat bottom microplates (TPPs, Trasadingen, Switzerland) were used to measure the growth kinetics of mono-species and binary *S. epidermidis* and *C. acnes* communities. Cultivation was established under anaerobic conditions in two different models. First, cultures of *C. acnes* and *S. epidermidis* were washed twice with sterile sterile physiological saline (PS—0.9% NaCl in distilled water) at 5000 g and 4 °C and adjusted to OD_540_ = 1 and 2 with sterile PS. Cultures with OD = 1 were used to study mono-species cultures, and cultures with OD = 2 were mixed in equal proportions to make dual-species communities. In the wells of the microplate, 200 μL of RCM with the addition of an appropriate amount of ANP per well was inoculated, and 10 μL of a cell suspension was inoculated into each well. Some wells were left without bacterial inoculation as empty controls. Wells without ANP were controls for ANP effect detection. Some wells with RCM without ANP were inoculated with 10 μL of OD_540_ = 1.7 overnight culture of strictly aerobic *Micrococcus luteus* to serve as an indicator of anaerobic conditions. *M. luteus* suspension was prepared in 15 mL of LB in a 50 mL conical flask at 33 °C and 150 rpm. All of the microplate wells were covered with 100 μL of sterile mineral oil as an additional barrier to oxygen. Wells on the perimeter of the plate were filled with 200 μL of sterile PS. The perimeter of the plate was covered with a 3 mm depth band of smooth modeling clay, and vacuum grease was applied over the clay. Clay and grease were used for hermetic isolation from atmospheric O_2_. In parallel, the hermetically closing bag (GasPack, BD) was sterilized with UV for 20 min. In the sterile bag, a gas-generating cartridge (Anaerogas) was placed immediately before plate incubation. The plate without a cover lid was installed into the special guiding frame, and the system was immediately placed in the prepared bag. Then, the cover lid of the plate was immediately placed into the bag to avoid contamination with grease and to leave the plate opened for O_2_ elimination. The bag was closed tightly and left for 2 h at RT to eliminate O_2_. After this incubation, the plate was tightly closed with the lid without unsealing the bag, the plate was then removed from the bag and the perimeter of the lid was covered with grease. The plate was incubated in the microplate spectrophotometer Xmark (Biorad) at 33.5 °C without shaking for 72 h. The reading was performed at 540 nm every 15 min. This model allowed us to study the mix of biofilms and planktonic cultures where the planktonic cultures dominated over the biofilm on the bottom of the well.

The second model potentially allows us to study the mix of biofilms and planktonic cultures where biofilms are predominant. Cultures of *C. acnes* and *S. epidermidis* were washed twice with sterile PS at 5000 g and 4 °C and adjusted to OD_540_ = 2 and 4 with sterile PS. Cultures with OD = 2 were used to study mono-species cultures, and cultures with OD = 4 were mixed in equal proportions to make dual-species communities. Two hundred microliters of the appropriate suspension was inoculated per well and left in an anaerobic atmosphere in a GasPack bag for 2 h to allow cells to adhere to the well bottom. Then, suspensions were removed from wells, wells were washed with 200 μL of sterile PS, and 200 μL of RCM medium with or without ANP was inoculated per well. Then, the *M. luteus* suspension was inoculated in some wells as described above. The wells were covered with sterile mineral oil, and the plate was prepared and incubated in GasPack as described above for 1 h. Then, the incubation and analysis of kinetics were performed in the same manner as described previously.

In the experiments, kinetic parameters of growth were analyzed—maximal growth rate (h^−1^), minimal generation time (h) and maximum OD_540_. We assumed the initial supposition that the OD of a culture is proportional to the cell amount in the suspension. Thus, the maximal growth rate was calculated by the formula µ = Ln((OD_2_/OD_1_)/(T_2_−T_1_)), where OD_1_ and OD_2_ are the OD at the beginning and end of the linear portion of the semilogarithmic plot of the growth curve, respectively. Semilogarithmic plots were built, and the linear portions were calculated using the function SLOPE in EXCEL. SLOPE allows us to find the coefficient “a” of linear regression y = ax + b; thus, we identified the portion as “linear” when it has both (i) minimal standard deviation of coefficient “a” and (ii) “a” is maximal—the curve is closer to vertical. Deviation time t was calculated using the formula t = ln2/µ.

### 2.8. Statistics and Data Processing

All experiments were conducted at least in triplicate. RNA extraction was performed in duplicate. Statistical analysis of data (except the RNA sequencing described above) was performed using the nonparametric Mann-Whitney U-test. All microbiological data plots were designed in Microsoft EXCEL 2007 Software. Where appropriate, average relative values (control without addition of ANP was designated as 100%), and absolute values were plotted on the graphs, and the standard error of the mean was depicted as error bars. Q-values were calculated from p-values in each experiment using false discovery rate correction, as proposed by Benjamini and Hochberg [27]. The q-values were marked on data plots.

## 3. Results

### 3.1. Effect of ANP on Biofilms on PFTE Cubes

First, we tested a wide range of ANP concentrations on *S. epidermidis* and *C. acnes* mono-species planktonic cultures and biofilms to find the most efficient concentration of the hormone. Because *C. acnes* grows slower than *S. epidermidis*, we incubated it for 72 h, while *S. epidermidis* was checked after 24 h and 72 h of incubation (established biofilms and long-term mature biofilms). We found that after 72 h of incubation, ANP significantly affected the biofilm growth of *C. acnes* without significant changes in planktonic growth (Figure 1A). The effect of ANP was dose-dependent: at the physiological level (6.5 × 10^−12^ M), it had a slight inhibitory effect on biofilms but with increasing concentration (up to 6.5 × 10^−9^ M), a strong stimulatory effect appeared, with a maximum at 6.5 × 10^−9^ M—163 ± 27% of the control. In the presence of 6.5 × 10^−10^ M ANP, the effect was also high—139 ± 18% of the control.

*S. epidermidis* was affected in a similar dose-dependent manner. Unlike *C. acnes*, planktonic cultures of *S. epidermidis* were more sensitive to ANP and after 24 h of incubation they were inhibited in the presence of physiological concentrations of ANP (88 ± 3% of control). This effect disappeared with increasing hormone concentrations (Figure 1B). After 72 h of incubation, the inhibitory effect of 6.5 × 10^−12^ M ANP (physiological) disappeared and changed to marginal stimulatory: the planktonic culture OD was 105 ± 3% in comparison with the control (Figure 1C). After 24 h of incubation, the increase in concentration led to the disappearance of the ANP effect. ANP had an inhibitory effect on biofilms after 24 h and 72 h of incubation. Further, after both time points, it was the strongest at 6.5 × 10^−10^ M (81 ± 5% of control and 65 ± 2% of control, respectively). Hence, for the next experiments, we chose a concentration of 6.5 × 10^−10^ M: at this concentration, the hormone had the opposite effect on the tested microorganisms, which made it interesting for bacterial dual-species community analysis.

### 3.2. Effect of ANP on Biofilms on GMFFs

In this model system, the biofilms grew on the surface of GMFFs without free liquid and hence without planktonic culture and the initial adhesion phase, as was the case on PFTE cubes. Additionally, to understand deeper processes in dual-species communities, we grew all mono-species and dual-species biofilms for 24 h and 72 h to compare short-term established and long-term established biofilms. We found that the CFU amount of *S. epidermidis* decreased significantly after 72 h of incubation in comparison with 24 h biofilms—in control samples, the CFU amount decreased from 1.1 × 10^10^ ± 4.1 × 10^9^ to 3.0 × 10^9^ ± 1.1 × 10^9^ per biofilm. ANP had no visible effect on the CFU count after 24 h of incubation but, after 72 h, it decreased the number of CFUs (to 1.9 × 10^9^ ± 8.8 × 10^8^ per biofilm, Figure 2A), but the error bars were too large to suggest that this result was statistically significant. Therefore, we could speak here only about a tendency not about the strictly confirmed difference. The metabolic activity of mono-species biofilms of *S. epidermidis* was also reduced after 72 h of incubation in comparison with 24 h (OD_540_ 4.1 ± 0.4 and 2.3 ± 0.6 in control samples after 24 h and 72 h, respectively, Figure 2B). Thus, after 72 h of incubation, biofilms of *S. epidermidis* were in the deep stationary phase of growth with reduced metabolic activity. There was no inhibitory tendency after 72 h of incubation. Instead, there was a tendency to stimulate MTT reduction by the addition of ANP after 24 h and 72 h of incubation (OD_540_ 4.7 ± 0.8 and 2.6 ± 0.6 after 24 h and 72 h in the presence of ANP, respectively, Figure 2B).

In contrast to *S. epidermidis*, *C. acnes* mono-species biofilms became better established after 72 h of incubation due to their slower growth in comparison with staphylococci (Figure 2C). In the control samples, the CFU amounts were 2.7 × 10^9^ ± 1.1 × 10^9^ and 1.4 × 10^10^ ± 2.7 × 10^9^ after 24 h and 72 h, respectively (5-fold increase during incubation). Metabolic activity in control samples was 3-fold lower in the 24 h samples in comparison with the 72 h mature biofilms: OD_540_ 0.95 ± 0.4 and 3.1 ± 0.3, respectively. The presence of ANP in the medium significantly increased the number of CFUs up to 4.5 × 10^9^ ± 6.7 × 10^8^ and 2.9 × 10^10^ ± 5.5 × 10^9^ per biofilm. The metabolic activity of *C. acnes* mono-species biofilms was also increased in the presence of ANP—slightly after 24 h (OD_540_ 1.1 ± 0.2) and significantly after 72 h (OD_540_ 4.7 ± 0.7). Thus, the effect of ANP on mono-species *S. epidermidis* and *C. acnes* on PFTE cubes was the same as that on GMFFs, and the ANP potentially changes some cellular systems that regulate processes independent of the initial adhesion stage and the surface type (the hydrophobic PFTE or the hydrophilic GMFFs). Additionally, in *S. epidermidis*, ANP stimulates the metabolic activity of cells at least at the level of the ability to reduce MTT (i.e., potential efficiency of reduced nicotinamide adenine dinucleotide (NADH) dehydrogenases).

In dual-species biofilms, the regulatory effect of 6.5 × 10^−10^ M ANP in RCM was especially clear (Figure 2E). The amount of CFU of *S. epidermidis* in control samples both in the 24 h and 72 h biofilms was higher than the CFU count of *C. acnes*: after 24 h of incubation the *S. epidermidis* CFU count was 1.68 × 10^10^ ± 5.3 × 10^9^, while *C. acnes* CFU count was 2.4 × 10^8^ ± 1.15 × 10^7^. Total CFU was 1.7 × 10^10^ ± 5.4 × 10^9^ per biofilm. Therefore, *S. epidermidis* dominated the dual-species community. In the presence of ANP, the amount of *S. epidermidis* CFU decreased to 8.4 × 10^9^ ± 4.7 × 10^8^. The *C. acnes* CFU increased to 3.1 × 10^8^ ± 1.7 × 10^7^. Therefore, if making proportions 1.68 × 10^10^: 2.4 × 10^8^ and 8.4 × 10^9^: 3.1 × 10^8^, ANP decreased the ratio of the *S. epidermidis—C. acnes* CFU count from 71:1 in the control to 27:1 in the presence of the hormone (in control and in presence of ANP it was 71 and 27 CFUs of *S. epidermidis* per 1 CFU of *C. acnes* respectively). In parallel, the measurement of metabolic activity showed a significant decrease in the MTT reduction level: OD_540_ was 4.3 ± 0.4 in the control and 2.2 ± 0.3 in the presence of the hormone (Figure 2F). This reduction is potentially the result of complex interactions between species in the biofilm because both showed no effect or stimulation of metabolic activity in the presence of ANP when cultivated alone.

After 72 h of incubation, the effect of ANP on dual-species communities was the opposite, as after 24 h and after 72 h, *C. acnes* became dominant instead of *S. epidermidis* in the number of CFUs. In the control samples, the number of CFUs of *S. epidermidis* and *C. acnes* were 1.0 × 10^8^ ± 4.7 × 10^7^ and 1.1 × 10^10^ ± 1.1 × 10^9^, respectively (Figure 2E). The total CFU amount was 1.12 × 10^10^ ± 4.8 × 10^9^. In the presence of ANP, the number of CFUs was as follows—for *S. epidermidis*, slight increase up to 1.5 × 10^8^ ± 9.1 × 10^7^ per biofilm; for *C. acnes*, a decrease to 4.1 × 10^9^ ± 2.3 × 10^8^ per biofilm; total CFU count, a decrease to 4.25 × 10^9^ ± 3.8 × 10^8^ per biofilm. Thus, if making a proportion, the ratio of the *S. epidermidis*—*C. acnes* CFU count in the presence of ANP increased from 1:110 in the control to 1:27 in the presence of the hormone. Because *C. acnes* dominated in the biofilm and its number of CFUs decreased, the metabolic activity of dual-species biofilms did not change in the presence of ANP (Figure 2F, (in mono-species biofilms of *C. acnes*, it was increased in the presence of the hormone)).

To better understand the potential mechanism of these changes in dual-species biofilms, we analyzed cell aggregation using light microscopy. Cell aggregates were considered as potential CFUs. *S. epidermidis* demonstrated a slightly increased aggregate size in the presence of ANP after 24 h of cultivation: 3.56 ± 0.8 cells per aggregate (CPA) and 4.1 ± 1.1 CPA, respectively (Figure 3A), but the ratio of aggregates and single cells increased from 12.4% in the control to 20.3% in the ANP samples (Figure 3B). Thus, most of CFUs here were single cells. After 72 h of incubation, no single cells were observed in the samples, and the cell aggregates were much larger than after 24 h: in controls, the average size of the aggregate was 116.8 ± 13 CPA and, in the presence of ANP, it decreased to 55.3 ± 7.1 CPA. Thus, long-term cultivation of *S. epidermidis* biofilms leads to the formation of large cell aggregates that are difficult to break up even with abrasive glass fibers of the GMFFs. Considering the decrease in CFU count and the decrease in cell aggregate size in mono-species *S. epidermidis* biofilms, we suggest that ANP decreases the ability of *S. epidermidis* to proliferate and form a biomass of cells but increases cell metabolism and MTT reduction. Additionally, ANP changes potential processes of cell aggregation (adhesion synthesis or matrix production).

In the case of *C. acnes*, the addition of ANP into the medium also significantly changed cell aggregation. After 24 h of incubation, in the control, the aggregation ratio was 15.6% (Figure 3D), and the average aggregate size was 5 CPAs (Figure 3C). The addition of ANP led to aggregation disappearance—no aggregates were found in *C. acnes* samples with ANP, and the CFUs were only single cells. After 72 h of incubation in mature biofilms in control samples, cell aggregates were easy to break up, and we suggest that all the *C. acnes* CFUs were single cells (Figure 3C,D). In the presence of ANP, the aggregation became more stable and the average aggregate size was 5.6 ± 1.5 CPA, and the aggregation ratio was 20.7%. Hence, remembering the increase in CFU count and increasing aggregation in mono-species biofilms of *C. acnes*, we suggest that ANP stimulates the growth of *C. acnes* cells in biofilms and potentially triggers some mechanisms of cell aggregation (adhesion synthesis or matrix production). Also, it can be a reason for *C. acnes* CFU count decrease in dual-species biofilms in presence of ANP.

We found that dual-species aggregates were a minor fraction of all aggregates in samples except control samples after 72 h of incubation, where only CFUs were found. In 24 h biofilms, we found no dual-species aggregates and all presented aggregates were mono-species aggregates of *S. epidermidis* with an average of 2.5 ± 0.5 CPA in size. The presence of ANP led to the occurrence of dual-species aggregates, which were 50% of all the aggregates in the samples (the rest were mono-species *S. epidermidis* aggregates) and 2.1% of all CFUs. The size of dual-species aggregates (DSA) was 2 ± 0.05 CPA of *S. epidermidis* and 3 ± 0.8 CPA of *C. acnes* (Figure 4A). Thus, *C. acnes* becomes at least an equal part of DSA in the presence of ANP in short-term established biofilms. The ratio of aggregation in the presence of ANP was not changed significantly and remained at only 4% in both the control and in the presence of ANP, while the total single cells comprised 96% of CFUs (Figure 4B). The number of *S. epidermidis* single cells decreased in the presence of ANP from 94% in the control to 79% of the total CFUs. The ratio of *C. acnes* single cells increased in the presence of ANP from 2% in the control to 16.5%. Thus, the increase in single cells can be a reason for the increase in the biomass ratio of *C. acnes* calculated from the CFU count after 24 h of incubation.

After 72 h of incubation, as in mono-species biofilms, aggregation was much stronger. As in mono-species biofilms, aggregation of *S. epidermidis* was the core process for CFU formation. In DSA, the amount of *S. epidermidis* was 38.0 ± 11 CPA in the control, while in the presence of ANP, it was reduced to 9.8 ± 2.8 CPA. *C. acnes* in DSA presented an average amount of 6.1 ± 2.8 in the control, and with ANP, its presence was reduced to 4.3 ± 3 CPA (Figure 4A). Mono-species aggregates (MSA) of both microorganisms were not found after 72 h of incubation in control samples, but in the presence of ANP, MSA of both *S. epidermidis* (77.8% of all aggregates) and *C. acnes* (7.4%) occurred, so the ratio of DSA was only 5.8% of all aggregates. The size of *S. epidermidis* MSA was 10.8 ± 6 CPA, while *C. acnes* formed aggregates of 6 ± 0.8 CPA. Additionally, the ratio of MSA was 4.6% and 0.4% of all CFUs for *S. epidermidis* and *C. acnes*, respectively, while single cells were 83.9% and 10.2% of the CFUs of *S. epidermidis* and *C. acnes*, respectively. Thus, the microscopy control of cell suspensions is on the one hand a useful tool to analyze potential cell aggregation, but on the other hand, it can lead to controversial results as occurred with CFU counting on a Petri plate and cell aggregation microscopy in the case of the 72 h biofilm without ANP. On Petri plates, there were CFUs of *C. acnes* and *S. epidermidis*, but on glass slides, there were only dual-species aggregates. This suggests that: (i) most of the *S. epidermidis* biomass in 72 h biofilms is located in rather strongly assembled aggregates that interfere with CFU counting on plates; (ii) these DSAs can probably be disrupted into smaller pieces during the plating of the suspension, and *C. acnes* cells can release from them and form colonies. Additionally, it seems difficult to differentiate *S. epidermidis* mono-species colonies and dual-species colonies because *S. epidermidis* grows faster than *C. acnes*, and it seems to have the same appearance as colonies in Petri dishes. Therefore, a deeper analysis of dual-species biofilms should be performed. Despite these circumstances, the ability of ANP to reduce the aggregation of *S. epidermidis* and, thus, of dual-species biofilms was clearly shown. Taking into consideration both the CFU count results and the aggregation analysis, we can suggest that, in this microbial community, *S. epidermidis* is a dominating bacterium, while *C. acnes* grows and becomes more present in the late stages of community formation.

### 3.3. CLSM Study of Biofilms

To better understand the structure of biofilms, we performed CLSM analysis of mono-species and dual-species biofilms labeled with FISH probes. We found that in this system, ANP stimulates the growth of *C. acnes* mono-species biofilms (Figure 5A and Figure 6). After 24 h of incubation, the average biomass density (ABD) was 0.58 ± 0.53 µm^3^/µm^2^ (Figure 6B) in the control and 1.7 ± 0.5 µm^3^/µm^2^ in the presence of ANP (Figure 6D). After 72 h of incubation, the ABDs were 1.1 ± 0.6 µm^3^/µm^2^ (Figure 6J) and 2.0 ± 0.4 µm^3^/µm^2^ (Figure 6L) in the control and in the presence of ANP, respectively. *S. epidermidis* was inhibited by ANP: after 24 h, its biomass density was 2.5 ± 0.5 µm^3^/µm^2^ (Figure 6A) and 0.75 ± 0.56 µm^3^/µm^2^ (Figure 6C) in the control and in the presence of hormones, respectively. After 72 h of incubation, the ABD of *S. epidermidis* (Figure 5B) mono-species biofilms was 1.9 ± 0.3 µm^3^/µm^2^ (Figure 6I) in the control and 0.35 ±0.05 µm^3^/µm^2^ (Figure 6K) in the presence of ANP. In dual-species biofilms, *C. acnes* was a minor component in this system. Its ABD was only 0.001 ± 0.00049 µm^3^/µm^2^ in control variants after 24 h of incubation (Figure 5A and Figure 6F). However, with the addition of ANP, it increased up to 0.0026 ± 0.0007 µm^3^/µm^2^, but it also remained very low (Figure 6H). After 72 h of incubation, the *C. acnes* ABD was 0.004 ± 0,003 µm^3^/µm^2^ (Figure 6N), while with ANP, it was 0.1 ± 0.023 µm^3^/µm^2^ (Figure 6P). Thus, the stimulatory effect of ANP was observed in both mono- and dual-species biofilms.

*S. epidermidis* in dual-species biofilms demonstrated interesting behavior. After 24 h of cultivation, *S. epidermidis* was slightly stimulated by *C. acnes* in samples without ANP in comparison with mono-species biofilms: its ABD was 3.1± 0.5 µm^3^/µm^2^ (Figure 5B, Figure 6E). ANP (6.5 × 10^−10^ M) significantly decreased the ABD of *S. epidermidis* to 0.82 ± 0.72 µm^3^/µm^2^ (Figure 6G). After 72 h of incubation, in control dual-species biofilms, *S. epidermidis* grew worse than in mono-species biofilms without ANP: its ABD was 1.3 ± 0.3 µm^3^/µm^2^ (Figure 6M). However, in the presence of ANP, in contrast to 24 h biofilms, *S. epidermidis* growth was stimulated up to 2.8 ± 0.6 µm^3^/µm^2^ (Figure 6O). Thus, the ratio of biomasses of the studied bacteria was changed in the presence of ANP and during the incubation time (Figure 5C). In 24 h biofilms, in the control and with ANP, the proportion of *C. acnes* was only 0.03% and 0.3% (10-fold increase), and after 72 h of incubation, it became 0.33 and 3.5%, respectively (the same 10-fold increase). Thus, ANP can change the balance between two bacteria in the community. It is especially interesting because in the system tested biofilms were grown in parallel with planktonic cultures, thus, *S. epidermidis* had a large handicap in terms of adhesion and growth because of its kinetic parameters. Even with such strong rival properties, *C. acnes* had the ability to grow, which was strongly increased with the addition of ANP.

### 3.4. Study of Kinetic Parameters of Planktonic Cultures and Biofilms

To further study the kinetic parameters of biofilms and planktonic growth of mono-species cultures and communities, we analyzed microbial growth using high-frequency OD_540_ measurement of bacterial biomass in wells of 96-well microtiter plates. First, we inoculated bacteria in liquid RCM with or without ANP as we did for the PFTE and CLSM experiments—a drop of cell suspension was diluted in RCM in the wells, so the biofilm on the bottom of the well was formed from the planktonic culture above. This is important because the beam of the scanning module travels vertically through the liquid and the bottom of the well; hence, we measured both biofilm and planktonic culture, where the latter had the dominant impact on the final OD. Thus, in this system (Figure 7A), *S. epidermidis* had an average maximal OD_540_ = 0.85 in both the control and in the presence of 6.5 × 10^−10^ M ANP, the minimal generation time (MiGT) was decreased from 2.86 h in the control to 3 h in presence of ANP and the maximal growth rate (MaGR) was decreased from 0.24 h^−1^ to 0.23 h^−1^ in presence of ANP. The linear portion of the growth curve (LPGC) occurred later and was reduced in the presence of ANP: in the control, it was 0.5–3 h from the start of incubation, and in the presence of ANP, it was 0.75–2.75 h from the start of incubation. The OD_540_ values after 24 h and 72 h of incubation were the same both in the control and in the presence of ANP, which correlates with the effect of ANP on planktonic cultures found on PFTE cubes. Thus, ANP had slight inhibitory activity on *S. epidermidis* at the beginning of culture growth in this system. *C. acnes* mono-species cultures were slightly stimulated—MiGT was decreased from 5.8 h in the control to 5.6 in the presence of ANP, and MaGR was increased from 0.11 h^−1^ in the control to 0.12 h^−1^ in the presence of ANP (Figure 7C). The LPGC occurred in the period 15.75–19 h in the control and 15–18.5 h from the start of incubation in the presence of ANP (longer in the presence of the hormone). However, the average maximal OD was slightly reduced to 1.19 in the presence of ANP (1.23 in the control). The OD_540_ of cultures after 72 h was the same in all samples, correlating to the results obtained in experiments with PFTE cubes. Thus, ANP was potentially able to accelerate the growth of a mono-species culture of *C. acnes* at the beginning of cultivation. The dual-species community had a general profile similar to a mono-species *S. epidermidis* culture (Figure 7E): its MiGT was 3.4 h in the control and 3.6 h in the presence of ANP (decrease), and the MaGR was 0.2 h^−1^ and 0.19 h^−1^ in the control and in the presence of the hormone, respectively. LPGC occurred at the beginning of the start of incubation in all samples, as in *S. epidermidis* mono-species cultures. In the control, LPGC occurred at 0.25–3.75 h from the beginning of the incubation, and in the presence of ANP, it occurred at 0.25–3.25 h (shorter than in the control). The OD of cultures was the same after 24 h and 72 h, correlating with planktonic growth in the system with PFTE cubes. Therefore, the parameters of the dual-species cultures were in between those of *S. epidermidis* and *C. acnes* mono-species parameters with a tendency to be closer to *S. epidermidis*.

The second system allows us to measure the OD of cultures with dominant biofilms because cells were initially adhered to the bottom of the well, and the planktonic culture formed from attached cells dispersed into the liquid during biofilm formation (Figure 7B,D,F). Here, we found that *S. epidermidis* showed different behavior in comparison with the first system—in the presence of the hormone, its MiGT was reduced to 3.1 h compared to 3.6 h in the control, and the MaGR was increased from 0.19 h^−1^ to 0.22 h^−1^ in the control and ANP samples, respectively. The LPGC was at 3–5 h and 3–5.5 h from the start of the experiment in the control and in the presence of the hormone, respectively. The maximal OD_540_ was 0.82 and 0.83 in the control and in the presence of ANP, respectively, but after 24 h of cultivation, the OD in the control was higher than that in the presence of ANP—0.68 and 0.64, respectively—and after 72 h, the OD was 0.63 and 0.56, respectively (Figure 7B). Thus, ANP potentially accelerated growth at the initial step of biofilm formation but decreased the amount of *S. epidermidis* biomass, as we found in previous experiments. The smaller difference may be explained by the presence of planktonic culture, which slightly affects the total OD of the sample.

In the case of *C. acnes*, we obtained results also correlating with previous results (Figure 7D). In the presence of the hormone, its MiGT was reduced to 6.7 h compared to 7.5 h in the control, and the MaGR was increased from 0.09 h^−1^ to 0.11 h^−1^ in the control and ANP samples, respectively. The LPGC was at 13.25–16.25 h and 13–15.75 h from the start of the experiment in the control and in the presence of the hormone, respectively (in the presence of ANP, it was shorter but potentially more intensive). The maximal OD_540_ was 0.88 and 0.95 in the control and in the presence of ANP, respectively. After 24 h of cultivation, the OD in the control was higher in the presence of ANP—0.67 and 0.72, respectively—and after 72 h the OD was 0.88 and 0.95 in the control and in the presence of ANP, respectively (actually, it is the maximal OD of the curves, Figure 7B). Thus, ANP stimulated the growth of *C. acnes* biofilms and, because of the slower growth rate, the difference between the control and ANP samples was more pronounced in comparison with *S. epidermidis*.

Dual-species communities generally behaved as the mean between *S. epidermidis* and *C. acnes* mono-species cultures, but they differed from dual-species communities in the previous system because of initial cell adhesion. The general character of the curves demonstrated a much higher dependence on ANP presence in the medium because of the smaller proportion of the planktonic culture of the total OD of the sample. The initial part of the curve was similar to the *S. epidermidis* curve because of its faster growth, but the later phase was closer to the *C. acnes* curves (Figure 7F). Additionally, ANP affected the growth of dual-species cultures, resulting in stimulation of growth. In the presence of the hormone dual-species MiGT, which was 3.6 h in both the control and ANP samples, the MaGR was 0.19 h^−1^ in all experiments. The LPGC was 3.25–5.25 h in the controls and in the presence of ANP. The maximal OD_540_ was 0.82 and 0.83 in the control and in the presence of ANP, respectively. After 24 h of cultivation, the OD in the control was higher in the presence of ANP—0.75 and 0.79, respectively—and after 72 h the OD was 0.68 and 0.73 in the control and in the presence of ANP, respectively. Thus, the amount of biomass was increased in the presence of the hormone, but kinetic parameters were the same, potentially due to the mixing of bacteria and the subsequent ANP effect superposition.

## 4. Discussion

The importance of regulating the microbial community on human skin has been of increasing interest in recent years. Continuing Lyte’s “microbial endocrinology” [2], the new concept of “Cutaneous bacterial endocrinology” emerged on the basis of different experimental data of host-bacterial interactions via peptidic hormones like NUPs, calcitonin gene-related peptide, substance P and so forth [28]. Human ANP seems to be a prospective agent of microbial homeostasis regulation on the skin, and this relationship could be a result of the long-term coevolution of humans and human microbiota. Indeed, it was shown previously that ANP can affect not only the mono-species cultures and biofilms of skin microbes but also their dual-species communities [9,10]. It was shown that ANP (and CNP) is able to change the balance between the more aggressive opportunistic pathogen *S. aureus* and the safer *S. epidermidis* when they are cocultivated in dual-species communities and also gives some benefits to *C. acnes* against *S. aureus*. This effect depended on the concentration of NUP and the cultivation conditions [9,10]. Moreover, some bacteria potentially have analogs of receptors of human NUPs [7]. In the present work, we expanded the studies of NUP effects in the case of ANP.

We found that ANP acts as a regulator of the bacterial community of *S. epidermidis* ATCC14990 and *C. acnes* RT5, allowing *C. acnes* to grow better in the presence of *S. epidermidis*, which is a dominant part of the microbial community. Additionally, it is important that ANP acts mostly on biofilms and less on planktonic cultures, which is evidence of a potentially evolutionary link between the multispecies biofilm community in skin glands and follicles and human regulatory systems. We did not examine the ANP effect under aerobic conditions in this study because of several reasons. First, we decided to start from the deeper microniches of skin follicles and sebaceous glands where anaerobic conditions occur and both two species contact with each other in biofilm. The second reason is that the tested strain of *C. acnes* is more sensitive to oxygen than others and does not grow in presence of oxygen. Thus, to make any correlation in dual-species communities in aerobic atmosphere would be more complex and require an additional methodology, which should be a point of a novel separate study. Also, in future studies the effect of NUPs should be investigated not only under normal atmospheric O_2_ concentration, but also in a range of smaller concentrations up to microaerophilic conditions due to the gradient of O_2_ in skin glands. According to the previous study, the effect of ANP was dependent on cultivation conditions [9,10]. Also, we decided to avoid the pre-treatment of cultures with ANPs before experiments just to make the initial test of ANP effect. The next step should be the deeper investigation of mechanisms of ANP action, and pre-treatment may be an efficient way to find some cellular changes in bacteria.

The slight effect on planktonic cultures is better observed in kinetic experiments with a microplate spectrophotometer and mostly occurs at the beginning of the growth curve with no effect on maximal OD (i.e., final biomass). The ability of ANP to give *C. acnes* an advantage may be a part of global regulatory interactions between human regulatory systems and skin (and other) microbiota. The effects of ANP on biofilms is complex, and all effects relate to global regulatory systems of cells because the effects do not depend on the surface type (hydrophobic PFTE, hydrophilic glass). The effects of ANP potentially lie in the area of cell-to-cell adhesion (decrease in self-adhesion of *S. epidermidis*, increase in adhesion of *C. acnes*) and matrix production, but this should be investigated further. Two major points during the investigation appeared to draw attention. First, CFU counting in the case of biofilms should be combined with other methods, such as microscopy control, because ANP can affect cell-to-cell aggregation and the CFU may be a single cell or a group of 100 cells. Another point is that during kinetic measurement in different microplate incubators, the dominant components of the suspension are important. Different types of cell preparations allow the creation of primarily planktonic or biofilm systems, and similar separations allow the results to be compared with each other more accurately. Concerning the aggregation of cells itself, it should be an important point in future investigation. Cell aggregation can be one of key targets of ANP action, because it seems to be that ANP has a multi-target manner of action, and potentially it affects more than 1 process in cells. This can also be an explanation of dose-dependence of ANP effect. Moreover, aggregation of bacterial cells to each other is a very complex process, nowadays even the new concept emerged of “the third lifestyle”—the non-surface attached bacterial aggregates [29]. These aggregates are encountered in many aqueous systems and in clinical situations (for instance in cystic fibrosis patients), thus, the ANP-mediated change of aggregation in *C. acnes* and *S. epidermidis* can be taken into consideration in future studies.

We showed for the first time that ANP can change the growth of *S. epidermidis* ATCC14990 and *C. acnes* RT5 (both isolated from human skin) at concentrations close to physiological blood plasma levels and can affect the mono-species and dual-species biofilms of these species. The impact of ANP on these species was recently studied using very high doses of the hormone (from 10^−8^ to 10^−6^ M), which are at least 100-fold higher than those we used in this study. Therefore, the present data are important for understanding the manner of action of ANP. In a previous study, it was shown that ANP at a concentration of 10^−8^ M decreased the growth of *C. acnes*, and in silico study shownt that the N-acetylmuramoyl-L-alanyl amidase of *C. acnes* presents 69% homology with the *P. aeruginosa* PAO1 amidase AmiC previously identified as the CNP receptor in this bacterial species [7]. But with a concentration decrease, it seems that this effect is generally changed to stimulation. Of course, we must always take into consideration that it is difficult to determine the actual concentration of any hormone on the skin, and especially inside the skin glands, where these bacteria form a community in anaerobic microniches [12]. Therefore, we used the normal concentration of ANP in blood plasma in our experiments. Additionally, we must take into consideration that hormones are potentially able to diffuse from blood plasma into the skin gland hollows and in the bulge area of hair follicles where the blood capillaries are in close proximity to the bacterial community inhabiting those sites. Additionally, hormones can potentially accumulate in such areas during sweat drying until high concentrations are reached in comparison with blood plasma. Thus, bacteria can potentially be affected by higher concentrations of hormones, including ANP. Of course, this hypothesis is speculative and needs to be approved or rejected through further experiments; however, our results argue in favor of this possibility. This is a potential reason why ANP affects the tested bacteria at concentrations higher than 6.5 × 10^−12^ M (normal physiological concentration in blood plasma). Additionally, another speculative hypothesis regarding why higher than physiological concentrations of ANP affect the pair *S. epidermidis*–*C. acnes* can be constructed. It was found that in pregnant women, the blood level of ANP is increased significantly (more than 2-fold, [23]). In parallel, we know that pregnancy is frequently associated with more severe acne vulgaris [30]; thus, potentially increased ANP levels could be one of the reasons for *C. acnes* stimulation and the worsening of acne vulgaris. Of course, this is a highly debatable hypothesis, especially in light of the fact that a previous article [9,10] presented another hypothesis of an ANP regulatory role in skin. According to that study, ANP is mostly an inhibitor of skin *S. aureus*, *S. epidermidis* and *C. acnes* [9,10] with the ability to give some advantages to safer bacteria against more dangerous bacteria. Additionally, there is information that patients with acne have a lower risk of heart diseases than patients with no acne [31,32], while high blood ANP levels can be a biomarker of cardiovascular disorders [33]. Here, we studied other concentrations of the hormone and found that the picture is much more complex. Moreover, the difference between ANP concentration in blood plasma is actually not significant between men (16.7+/−10.0 pg/mL) and women (18.8+/−11.7 pg/mL) [34]; however, it seems that adult women have acne vulgaris more frequently than men [35]. Probably, this fact can be explained by more complex situation in skin and potentially not only ANP is an effector of worsening acne vulgaris. For instance, catecholamines can change the ability of *C. acnes* to stimulate sebum synthesis in sebocytes [36]. Thus, skin bacteria should be affected by various chemical factors simultaneously, and the effect of ANP may be modified and interfered with by other compounds which differ in their concentrations between men and women. Also, the ANP is a peptide with small half-life in blood—about 2–15 min [37], endopeptidases are able to cleave it in blood [38]. In skin, it may be presented continuously, but it should be investigated. Thus, its effect in longtime incubation periods (24 h and more) may be a consequence of either some crucial metabolic changes in the very beginning of cell incubation, or some other mechanism which should be investigated further. The question of ANP dose dependent action must also be further evaluated. The probable answer here can be a multi-target action when multiple genes change their expression in different ways in different concentrations, which is why the effect of the same compound in different doses can be different and difficult to predict. Additionally, as was mentioned above, we must remember that, when we extrapolate experimental data on real skin and make hypotheses, NUPs other than ANP can accumulate on the skin as active compounds. Other hormones, blood components, sweat, minerals, microbial metabolites, and so forth can accumulate in sweat, sebaceous glands and hair follicles. These compounds may have a very different impact on skin microbiota homeostasis; hence, we are now just at the beginning of understanding host–bacteria interactions. Regardless, the ANP effect on bacterial biofilms and dual-species communities provides a good foothold for further investigations, giving wide perspectives in this area, including interest for dermatology and cosmetology. Acne vulgaris is a common skin disorder which in 2015 was detected in more than 600 million people in the world [39]. Different studies have aimed to find out ways to eradicate it. For instance, different natural and artificial compounds of different origins have been examined for anti-acne effect. For example, cinnamon oil [40], essential oils of *Myrtus communis* and *Origanum vulgare* in combination with the artificial retinoid Tretinoin [41] were shown to be effective anti-acne agents. Sometimes the positive anti-acne activity of such compounds may be a result of indirect changes in the humoral system. Actually, androgen secretion and following acne development were reduced in the golden hamsters model by use of the juice and essential oils of bergamot and sweet orange [42]. Thus, it is potentially possible to somehow modify the ANP action via novel cosmetics and reduce acne vulgaris.

## 5. Conclusions

We have shown the regulatory effect of ANP on dual-species biofilms of *C. acnes* RT5 and *S. epidermidis* ATCC14990. The present study adds to the body of evidence on how NUPs regulate human skin microbiota. As in a previous study [9], *C. acnes* competitive properties against staphylococci were stimulated in the presence of ANP, which resulted in an increased biomass ratio of *C. acnes* in dual-species biofilms. Also, the character of dual-species communities depended on the culture preparation (mostly planktonic or biofilm cultures) and, in both systems, *C. acnes* was further stimulated in the presence of the hormone. Hence, it is possible that NUPs and *C. acnes* are interconnected deeper than suggested before and the deeper investigation of NUPs, and particularly ANP, molecular mechanisms of action are needed to shed light on bacteria-NUPs interactions.

## Figures and Tables

**Figure 1 microorganisms-09-00552-f001:**
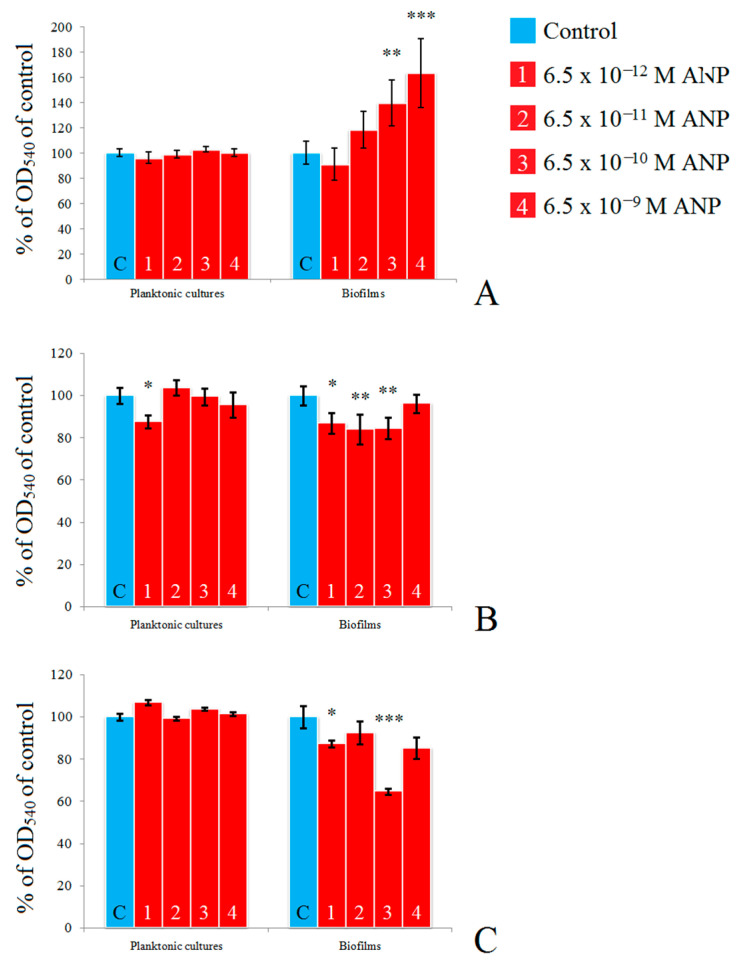
Effect of different concentrations of atrial natriuretic peptide (ANP) on *C. acnes* and *S. epidermidis* mono-species planktonic cultures and biofilms on on polytetrafluoroethylene (PTFE) cubes. Biofilms were stained with CV. (**A**)—*C. acnes* 72 h; (**B**)—*S. epidermidis* 24 h; (**C**)—*S. epidermidis* 72 h. The absence of asterisks means q > 0.05, * means q < 0.05, ** means q < 0.005, *** means q < 0.001.

**Figure 2 microorganisms-09-00552-f002:**
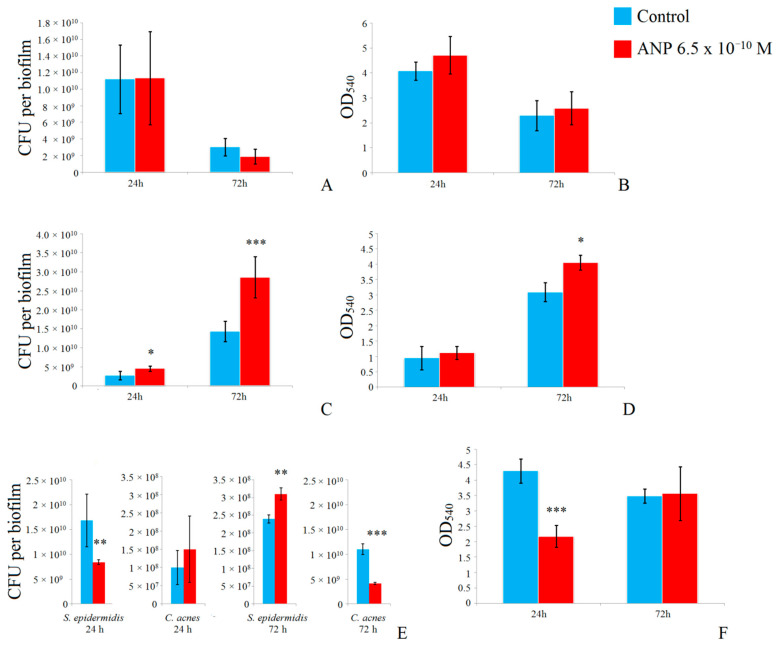
Effect of 6.5 × 10^−10^ M ANP on mono-species and dual-species biofilms of *S. epidermidis* and *C. acnes* on glass microfiber filters (GMFFs). (**A**)—colony forming unit (CFU) count of *S. epidermidis* mono-species biofilms; (**B**)—metabolic activity of *S. epidermidis* mono-species biofilms; (**C**)—CFU count of *C. acnes* mono-species biofilms; (**D**)—metabolic activity of *C. acnes* mono-species biofilms; (**E**)—CFU ratio in dual-species biofilms; (**F**)—metabolic activity of dual-species biofilms. Average means were plotted on the figure. The absence of asterisks means q > 0.05, * means q < 0.05, ** means q < 0.005, *** means q < 0.001.

**Figure 3 microorganisms-09-00552-f003:**
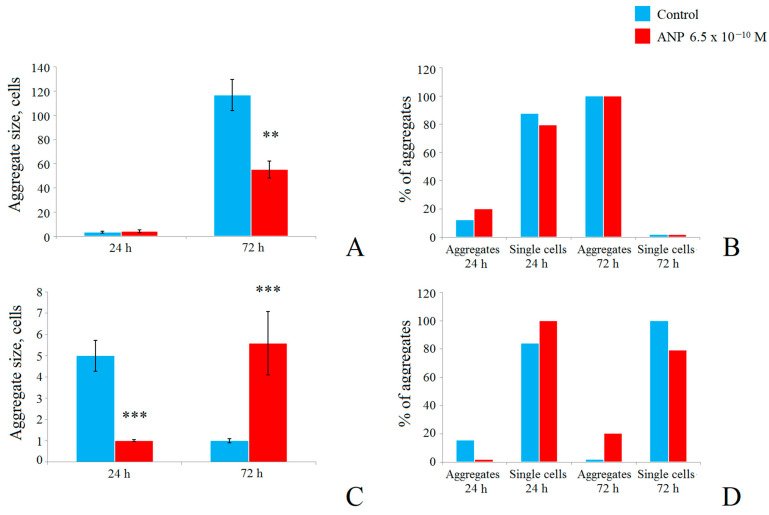
Analysis of aggregation in mono-species and dual-species biofilms of *S. epidermidis* and *C. acnes* on GMFFs. (**A**)—*S. epidermidis* aggregate size in mono-species biofilms; (**B**)—*S. epidermidis* aggregate ratio in mono-species biofilms %; (**C**)—*C. acnes* aggregate size in mono-species biofilms; (**D**)—*C. acnes* aggregate ratio in mono-species biofilms %. The absence of asterisks means q > 0.05, ** means q < 0.005, *** means q < 0.001.

**Figure 4 microorganisms-09-00552-f004:**
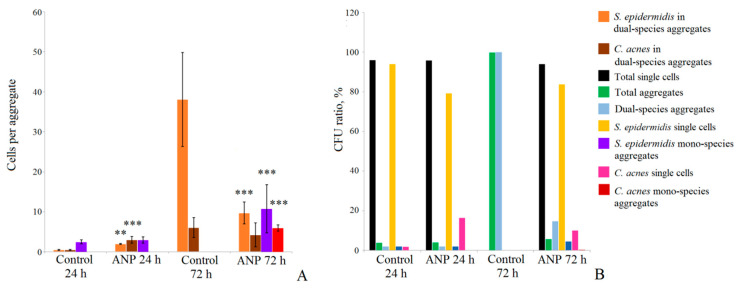
Composition and ratio of aggregates in dual-species biofilms. (**A**)—aggregate size; (**B**)—aggregation ratio. The absence of asterisks means q > 0.05, ** means q < 0.005, *** means q < 0.001.

**Figure 5 microorganisms-09-00552-f005:**
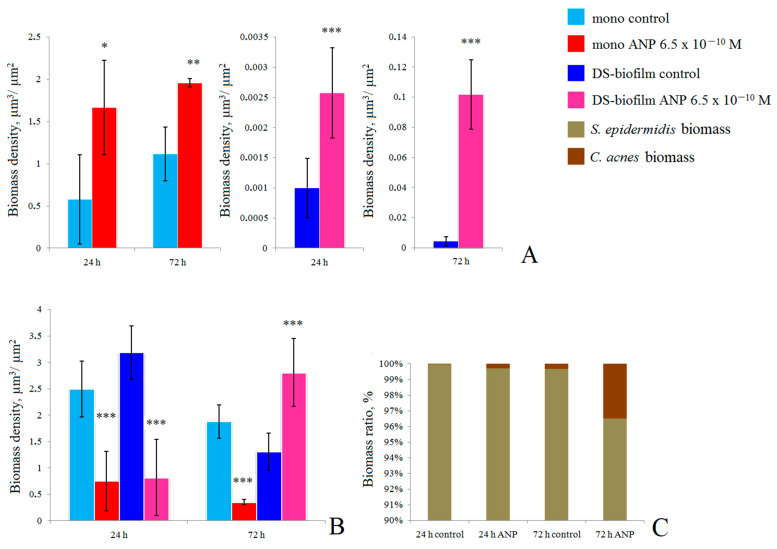
Average biomass density in mono-species and dual-species biofilms labeled with Fluorescent In Situ Hybridization (FISH) probes. (**A**)—*C. acnes*; (**B**)—*S. epidermidis*; (**C**)—Biomass ratio. The absence of asterisks means q > 0.05, * means q < 0.05, ** means q < 0.005, *** means q < 0.001.

**Figure 6 microorganisms-09-00552-f006:**
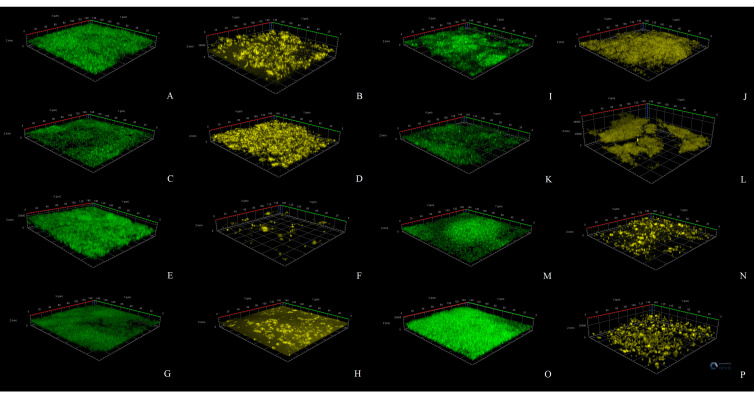
CLSM 3D-image visualization of *S. epidermidis* and *C. acnes* mono- and dual-species biofilms. *S. epidermidis* was stained with a fluorescein amidite (FAM)-containing FISH probe, and *C. acnes* was labeled with an R6G probe. (**A**)—*S. epidermidis* in 24 mono-species control biofilms; (**B**)—*C. acnes* in 24 mono-species control biofilms; (**C**)—*S. epidermidis* in 24 mono-species biofilms in presence of ANP; (**D**)—*C. acnes* in 24 mono-species biofilms in presence of ANP; (**E**)—*S. epidermidis* in 24 dual-species control biofilms; (**F**)—*C. acnes* in 24 h dual-species control biofilm; (**G**)—*S. epidermidis* in 24 h dual-species biofilm in presence of ANP; (**H**)—*S. epidermidis* in 24 h dual-species biofilm in presence of ANP; (**I**)—*S. epidermidis* in 72 h mono-species control biofilm; (**J**)—*C. acnes* in 72 h mono-species control biofilm; (**K**)—*S. epidermidis* in 72 h mono-species biofilm in the presence of ANP; (**L**)—*C. acnes* in 72 h mono-species biofilm in the presence of ANP; (**M**)—*S. epidermidis* in 72 h dual-species control biofilm; (**N**)—*C. acnes* in 72 h dual-species control biofilm; (**O**)—*S. epidermidis* in 72 h dual-species biofilm in the presence of ANP; (**P**)—*C. acnes* in 72 h dual-species biofilm in the presence of ANP.

**Figure 7 microorganisms-09-00552-f007:**
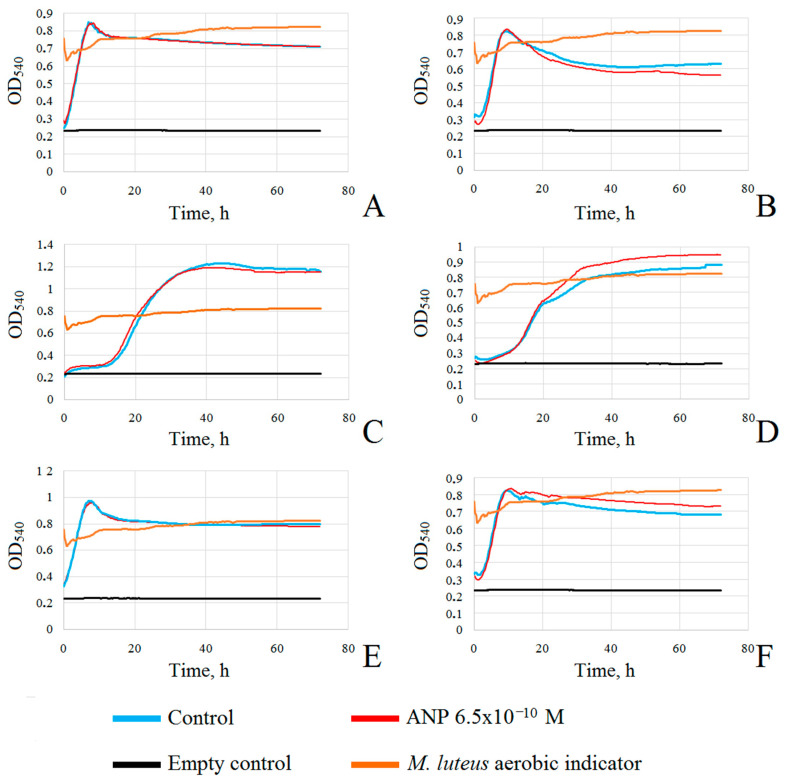
Growth curves of mono-species and dual-species cultures of *S. epidermidis* and *C. acnes*. (**A**)—mono-species cultures of *S. epidermidis* in planktonic culture domination; (**B**)—mono-species cultures of *S. epidermidis* in biofilm domination; (**C**)—mono-species *C. acnes* cultures in planktonic culture domination; (**D**)—mono-species *C. acnes* cultures in biofilm domination; (**E**)—dual-species cultures in planktonic culture domination; (**F**)—dual-species cultures in biofilm domination.

## Data Availability

Not applicable

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
