# Peer review of "Atrial Natriuretic Peptide Affects Skin Commensal Staphylococcus epidermidis and Cutibacterium acnes Dual-Species Biofilms"

_microorganisms, 2021, doi:10.3390/microorganisms9030552_

Round 1

Reviewer 1 Report

In the current study the author provided significant insight into the effect of ANP on dual-species  biofilms of C. acnes RT5 and S. epidermidis ATCC14990. Hence, it is possible that NUPs and C. acnes interaction may provide deeper insight into how human neuroendocrine systems can regulate the cutaneous microbial community and should be important for future applications in the fields of dermato-and cosmetology. This article is well designed, clear and well written.

Minor Comments:

The mean ANP value in healthy adult subjects of both sexes was 17.8+/-10.9 pg/ml with no significant difference between men (16.7+/-10.0 pg/ml) and women (18.8+/-11.7 pg/ml). However, an adult women experience acne at higher rates than their male (https://www.ncbi.nlm.nih.gov/pmc/articles/PMC5300732/ ). How author correlates this based on current study, “ANP which resulted in an increased biomass ratio of C. acnes in dual-species biofilms”

In the current study, the potential effect of NUPs on C. acnes may be supported by the observation that the N-acetylmuramoyl-L-alanyl amidase of C. acnes presents 69% homology with the P. aeruginosa PAO1 AmiC previously identified as the CNP receptor in this bacterial species. This study from authors previous research should be included in the discussion part of the manuscript.

Higher resolution images for Figure 3 and 4 should be added in the current manuscript

Author Response

Dear Reviewer,

Thank you for your comments. We have taken them into considetation and modified the text accorfing to them. Please, find the answers into the attached file.

Best regards,

Andrei

Reviewer 2 Report

General comments:

The present manuscript provides highly valuable results on the effects of a host neurohormone, atrial natriuretic peptide (ANP), on essential cutaneous bacteria, S. epidermidis and C. acnes, grown alone or in association in planktonic or sessile cultures. The techniques employed are innovative and allow an evaluation of the bacterial behavior in situations close to the physiological conditions, when an equilibrium can establish between free and adherent microorganisms. Some details on the protocol and data analysis are necessary. Results are abundant but sometimes difficult to analyze, both because the existence of limited variations and the need of figures quality improvement. The manuscript is well written, but some chapters need to be clarified as detailed elsewhere.

Point-by-points comments:

1 – A general checkup of the figures is necessary. Legends are excessively small and almost impossible to read, particularly in Figures 2, 3 and 4. As indicated in the legends to the figures, significance levels are normally indicated by asterisks, but these marks are lacking in Figures 2, 3 and 4B. That is making complicated to appreciate the results since in addition some differences are minimal.

2 – Line 158: It is mentioned that “both microbes were cultivated in anaerobic conditions”. We can presume that was the case for all other studies but that needs to be indicated. As the switch from aerobic to anaerobic conditions has a major impact on the bacterial metabolism, this should be also indicated in the abstract and discussed in the corresponding chapter.

3 – Chapter 2.3 : It should be necessary to indicate clearly if, for ANP treated bacteria, the pre-culture was realized in the absence or presence of the peptide as this can have an influence on the initial adhesion behavior. Were the pre-cultures also realized in anoxic condition.

4 – Chapter 2.4.: It should be also necessary to indicate for which time bacteria were exposed to ANP before layering on GMFF filters. This is essential since the concentrations of ANP employed were low and this peptide can bind on glass, particularly considering the high developed surface of glass filters.

5 – The sentence in line 410 is not coherent with the data presented in Figure 1B. In planktonic culture we see a marginal decrease under the effect of ANP in S. epidermidis, whereas the peptide has no effect on C. acnes, and a decrease of biofilm formation (except at 6.5 x 10-9 M), whereas ANP stimulated C. acnes biofilm formation.

6 – Lines 469to 481: Figure 2E is difficult to read and should be modified to visualize more clearly the changes in measured UFC. The text indicates important (significant?) variations, which are not visible on the figure.

7 – Lines 507-508 the sentence is confusing, it is unclear how were calculated the ratio 1:110 and 1:27.

8 – The discussion can be improved by considering the complex relations between cutaneous peptides and the skin microbiota. It should be interesting to mention the article in Exp. Dermatology 29(9):1-11, 2020. doi : 10.1111/exd.14158 introducing the concept of “cutaneous microbial endocrinology”.

9- The impact of ANP on the aggregation worth being also discussed in details. Aggregation is a phenotype shared by many bacterial species but rarely investigated by itself. As we recently observed (unpublished data) this phenotype can be differentially regulated in regard of the planktonic and biofilm modes of development suggesting that it could be considered as another bacterial mode of life, presumably associated to specific inter-bacterial links.

10 – The discussion on the local concentration of ANP in skin is interesting but I am not convinced by the hypothesis about peptide accumulation developed in lines 848 to 852 and 880-881. Indeed, ANP is a labile peptide with short half-life in blood (<1 min). It is spontaneously degraded by metalloproteases which are present on almost all cell membranes. Then, its effect on a bacterial populations can result either from an epigenetic transfer of the information through bacterial generations (memory effect) or from a continuous production. Many NUPs exist and some, such as BNP, can be released locally in high concentrations by nerve terminals (the concentration of neurotransmitters in a synaptic cleft can be over10-6 M).

Author Response

Dear Reviewer,

Thank you a lot for constructive and useful comments. We have taken them into account and modified the text in accordance with them. Please, find our answers in the attached file.

With best regards,

Andrei

Reviewer 3 Report

Dear authors 

ANP reduces the growth rate of S. epidermidis biofilms and increases that of C. acnes. The effect of ANP does not depend on the type of surface and probably affects other targets in microbial cells. Therefore, the potential regulatory effect of human ANP on dual species communities of skin microbes has been demonstrated and its potential has been shown to modify microbiota homeostasis on the skin.

Analysis by paper partitions:

1 - Introduction: must be reformed in the content and in the writing of the general part review the syntax of the topic

2- Discussion :Deepen the discussion regarding the molecules of natural origin with antimicrobial activity and in particular in the acne pathology are having a good result in clinical application. Learn more about this aspect using and citing the following references:

PMID: 32210603 ; PMID: 32587816 ; PMID: 33082712

3 - Check the bibliographic entries throughout the text, some of which are non-compliant, review some entries in the references and necessarily insert those referred to in point 2 for the purpose of acceptance by me.

4 - Review the English grammar and in particular the applied scientific English: in particular, verbal tenses and syntax in the discussion.

Author Response

Dear Reviewer,

Thank you a lot for your useful comments and notices. Please, find the attached file with our answers. We have taken into consideration your remarks and modified the text.

With best regards,

Andrei
